# Circulating miRNA Correlates with Lipid Profile and Disease Activity in Psoriatic Arthritis, Rheumatoid Arthritis, and Ankylosing Spondylitis Patients

**DOI:** 10.3390/biomedicines10040893

**Published:** 2022-04-13

**Authors:** Krzysztof Bonek, Ewa Kuca Warnawin, Anna Kornatka, Magdalena Plebańczyk, Tomasz Burakowski, Włodzimierz Maśliński, Małgorzata Wisłowska, Piotr Głuszko, Marzena Ciechomska

**Affiliations:** 1Department of Rheumatology, National Institute of Geriatrics, Rheumatology and Rehabilitation, 02-637 Warsaw, Poland; malgorzata.wislowska@spartanska.pl (M.W.); zruj@mp.pl (P.G.); 2Department of Pathophysiology and Immunology, National Institute of Geriatrics, Rheumatology and Rehabilitation, 02-637 Warsaw, Poland; ewa.kuca-warnawin@spartanska.pl (E.K.W.); anna.kornatka@spartanska.pl (A.K.); magdalena.plebanczyk@spartanska.pl (M.P.); tomasz.burakowski@spartanska.pl (T.B.); wlodzimierz.maslinski@spartanska.pl (W.M.)

**Keywords:** miRNA, psoriatic arthritis, rheumatoid arthritis, ankylosing spondylitis, inflammation, lipids, biomarkers, rheumatic diseases, lipid profile

## Abstract

This study aimed to investigate the associations of microRNA (miRs) signatures with cytokines, serum lipids, and disease activity in patients with psoriatic arthritis (PsA), ankylosing spondylitis (AS), and rheumatoid arthritis (RA). In total, 65 patients (PsA *n* = 25, AS *n* = 25, RA *n* = 15) and 25 healthy controls (HC) were enrolled into the study. The expression of miR-223-5p, miR-92b-3p, miR-485-3p, miR-10b-5p, let-7d-5p, miR-26a-2-3p, miR-146b-3p, and cytokines levels were measured in sera. DIANA-mirPath analysis was used to predict pathways targeted by the dysregulated miRs. Disease activity scores were calculated. Lipid profile, uric acid, glucose level, and C-reactive protein (CRP) concentrations were determined in the blood. Based on lipid profiles, the PsA group had hypertriglyceridaemia, and RA patients revealed mixed dyslipidaemia, while in AS, no specific changes were found. miR expression analysis revealed upregulation of miR-26a-2-3p and miR-10b-5p in PsA, miR-485-3p in AS, and let-7d-5p in RA. Several correlations between disease activity indexes, metabolites levels, and expression of miRs were observed in PsA, RA, and AS patients. Finally, in ROC analysis, miR-26a-2-3p/miR-485-3p, and let-7d-5p/miR-146b-3p tandems revealed high sensitivity and specificity in distinguishing between PsA, AS, and RA. Our study illustrates the superiority of miR expressions in distinguishing between RA, PsA, and AS. In PsA, a unique regulatory pathway exists through miR-26a-2-3p, miR-223-5p, miR-10b-5p, and miR-92b-3p that converges proatherogenic metabolism and disease activity.

## 1. Introduction

Rheumatoid arthritis (RA) and spondyloarthropathies (SpAs) are chronic inflammatory conditions that are associated with systemic inflammation and increased cardiovascular (CV) risk [1]. RA is a heterogeneous, autoimmune disease associated with symmetrical joint inflammation, leading to the destruction of bone and cartilage [1,2]. Ankylosing spondylitis (AS) and psoriatic arthritis (PsA) are classified as autoinflammatory forms of SpAs, but as opposed to RA, AS affects mostly the axial skeleton, entheses, and large asymmetrical joints [3,4]. PsA is a form of “psoriatic disease” with multiple phenotypes that can potentially affect axial and peripheral joints, as well as entheses. The differences in clinical images seem to be overlapping with different pro-inflammatory and cardiovascular (CV) risk profiles [5,6]. Despite differences in clinical images and comorbidities, distinguishing RA, AS and PsA in the stage of “early arthritis” is still considered challenging in clinical practice [7]. Research is ongoing on the utility and specificity of serological markers in the diagnosis or prognosis of rheumatic diseases. Autoantibodies are considered to be useful in patients with early arthritis, but almost 30% of patients positive for rheumatoid factor (RF), anticitrullinated peptide antibodies (ACPAs), or anti-carbamylated peptide antibodies (CARB) do not develop RA [8] and 40% of RA cases are seronegative [9]. In certain cases, the diagnosis could be established based on changes in X-ray, but in RA, only 60% of patients develop bone erosions after 1 year of disease progression [10]. In reality, the ideal serological biomarker panels for PsA and AS do not exist; therefore, there is a strong need to identify the new factors which allow distinguishing early RA from PsA and AS. MicroRNAs (miRs) have been shown to participate in the regulation of almost every cellular process. miRs constitute a large family of small, approximately 21-nucleotide long, non-coding RNAs. They are involved in posttranscriptional gene silencing through their pairing to the target mRNA, subsequently resulting in protein synthesis inhibition. In addition, dysregulated miRs have been shown to be linked to numerous diseases including rheumatic diseases. Therefore, miRs may be utilised as diagnostic and prognostic biomarkers. Our studies revealed that pro-inflammatory miR-146b plays a role in RA progression, whereas the level of miR-5196 is increased in AS, RA, and systemic sclerosis (SSc) patients [11]. Interestingly, changes in miR-5196 expression following 6 months of TNF inhibition treatment were a better biomarker for monitoring therapy response than the changes in the CRP level, in both RA and AS patients [11]. In addition, various circulating miRs have been shown to contribute to lipid metabolism alteration resulting in increased CV risk profile and pro-inflammatory cytokines production in autoimmune diseases, suggesting an important role of miRs as biomarkers for monitoring early diagnosis or treatment outcomes [4,8]. Cytokines are small proteins that regulate immune processes. It is assumed that an imbalance between pro-inflammatory and anti-inflammatory cytokines plays a crucial role in inflammation and the development of rheumatic diseases. TNF and IL-6 have central roles in the development and perpetuation of RA and lipid disturbances that are considered a “lipid paradox” [12]. IL-17 is critically involved in the pathogenesis of AS, and PsA, as well as altered metabolic profile in PsA [13]. Pro-inflammatory cytokines are involved in disturbed lipid metabolism and hence with a risk of CV disease development. Nevertheless, the link between disease activity, specific miR profiles, and increased CV risk in RA and SpAs seem to be complex. Thus, understanding the pathways by which miRs may connect clinical symptoms and systemic inflammation could be a new promising direction in further studies. Our study aimed to evaluate the use of miRs in differentiating RA from PsA and AS and to determine the associations between altered metabolic profile and systemic inflammation in RA and SpAs.

## 2. Materials and Methods

### 2.1. Patients

A total of 65 patients (15 with RA, 25 with PsA, 25 with AS), and sera from 25 healthy controls from a regional blood centre (HC) were enrolled in the study. All patients gave their written consent to participate in the study. The study was approved by the Ethics Committee of the National Institute of Geriatrics, Rheumatology, and Rehabilitation (Approval No. KBT-6/2/2019). PsA diagnosis was based on Classification Criteria for Psoriatic Arthritis (CASPAR) [14]. AS diagnosis was carried out based on Assessment of Spondyloarthritis International Society (ASAS) criteria for axial spondyloarthropathy with X-ray changes [3]. RA was diagnosed based on ACR/EULAR criteria [15] and at least three erosions in three different localisation in X-rays of hands and feet [16]. Exclusion criteria were diagnosis of other rheumatic diseases, acute and chronic infections, and a history of malignancies. Sera were obtained from peripheral blood samples during routine procedures after night fasting. Total cholesterol (TC), triglycerides (TG), low-density cholesterol (LDL), high-density cholesterol (HDL), C-reactive protein (CRP), pro-inflammatory cytokines levels, and miR expression were determined. All patients underwent physical examination, and disease activity was measured with the Disease Activity index in Psoriatic Arthritis (DAPSA) and Composite Psoriatic Disease Activity Index (CPDAI) in PsA. All PsA patients were diagnosed with axial and peripheral joint involvement, thus according to guidelines [17], disease activity was also measured by DAS28-CRP and ASDAS-CRP for all patients. In AS, disease activity was measured with Ankylosing Spondylitis Disease Activity Score (ASDAS-CRP) and with Disease Activity Score-28 (DAS28-CRP) in RA. All scores were calculated with standard formulas.

### 2.2. Circulating miR Profiling

The volume of 300 µL of sera from PsAs, AS, RA and HC was used to isolate circulating miRs using NucleoSpin^®^ miRNA plasma/serum (Macherey–Nagel, Düren, Germany), according to the manufacturer’s protocol. miR was extracted with 40 uL of ultra-pour water. TaqMan^®^ microRNA RT Kit was used to reverse transcribe to cDNA with the use of TaqMan^®^ MicroRNA Assays for hsa-miR-222-3p (471527_mat), hsa-let-7d-5p (000377), hsa-miR-223-5p, hsa-miR-146b-3p, hsa-miR-10b-5p, hsa-miR-26a-2-3p, hsa-miR-92b-3p, and hsa-miR-485-3p. Based on previously published data, miR-16-5p (477860_mir) was used to normalise circulating miR expression [11]. TaqMan^®^ miRNA Assays and TaqMan^®^ Universal PCR Master Mix II no UNG were used to quantify miR expression by quantitative polymerase chain reaction (qPCR) using QuantStudio 5 qRT-PCR machines (all from Thermo Fisher Scientific, Waltham, MA, USA). The expression levels of miRs relative to the average HC (arbitrarily set at 1) were calculated using the following equation: 2^−ΔΔCT^.

### 2.3. Statistical Analysis

Statistical analysis was performed using the SPSS module version 27 PS IMAGO. Alpha was set at 0.05. Shapiro–Wilk test for normality was performed for all parameters. Data are shown as mean with standard deviation (SD) for normal ranges and as medians with interquartile range (IQR) for non-normal ranges. Pearson’s test for normal ranges and Spearman’s correlation test for non-normal ranges were performed. For comparing numerical data, we performed Student’s *t*-test for normally distributed data and Mann–Whitney test for non-normally distributed data. Normally distributed data were used in linear and multivariate regression models. The threshold for ROC was calculated based on Youden Index.

### 2.4. KEGG Enrichment Analysis

Enrichment analysis of selected miRs was performed using DIANA-mirPath software. This algorithm was used to predict the interaction between miR and its mRNA targets based on the coding sequence (CDS). Illustrated interactions were previously filtered by FDR correction, a *p*-value threshold of less than 0.05, and Fisher’s exact test. 

### 2.5. ELISA

Sera were isolated from blood by routine laboratory methods, and samples were stored in aliquots at –70 °C until assayed. Concentrations of IL-17, TNF, and IL-21 were analysed by respective ELISA Ready-SET-Go Kits (eBioscience, San Diego, CA, USA), according to the manufacturer’s instructions. The concentration of IL-6 was analysed as described previously [18]. 

## 3. Results

### 3.1. Clinical and Demographic Data

Patients with RA, PsA, and AS were comparable in terms of age, sex, and disease duration. The distribution of male to female was similar to populational data, as RA mostly affects women, AS mostly affects men, and in PsA, the male-to-female ratio is described as equal. Clinical and demographic data are given in Table 1. 

All groups presented high clinical disease activity (mean DAPSA: 38.5; ASDAS-CRP: 3.7; DAS28: 5.98). Patients with RA showed a tendency towards higher CRP levels and had higher ESR than patients with PsA and AS. Patients with RA revealed mixed dyslipidaemia and higher fasting glucose levels, PsA patients had hypertriglyceridaemia, and in AS, no specific lipid profile was found (Table 1). A summary of applied treatment is presented in Table 2.

### 3.2. Validation of Selected miR Candidates and KEGG Enrichment Analysis

Based on previously published data highlighting the important role of miRs in lipid metabolism and autoimmunity, we measured the level of selected miR candidates—namely, miR-222-3q, let-7d-5p, miR-223-5p, miR-146b-3p, miR-10b-5p, miR-26a-2-3p, miR-92b-3p, and miR-485-3p—in PsA, AS, and RA sera. In PsA sera, the levels of miR-10b-5p (*p* = 0.0034), miR-26a-2-3p (*p* < 0.0001), and miR-485-3p (*p* = 0.0011) were upregulated. Similarly, circulating miR-26a-2-3p (*p* < 0.0001) and miR-485-3p (*p* < 0.0001) were also increased in AS sera (Figure 1).

In Figure 1, it can be seen that the levels of let-7d-5p (*p* = 0.0001), miR-146b-3p (*p* = 0.0035), miR-485-3p (*p* = 0.0465) were significantly increased, whereas miR-92b-3p (*p* = 0.0321) was reduced in RA sera, compared with HC sera. To find the target genes of selected miRs, we used KEGG enrichment analysis using Diana-mirPath v.3 software. 

In Figure 2A,C, it can be seen that four dysregulated miRs in RA and AS sera are involved in ECM–receptor interaction, among other pathways. Indeed, the bioinformatics analysis predicted that miRs altered in RA are involved in the negative regulation of different types of collages (COL1A1, COL1A2, COL3A1, COL5A1, and COL27A1) and integrins (ITGA8, ITGA8, ITGA8, ITGA8, and ITGA8) (Figure 2A). Similarly, selected miRs (-26a-2-3p, -485-3p) in AS sera can also regulate ECM–receptor signalling pathways via COL6A2 and thrombospondin 1 (THBS1) inhibition (Figure 2C). Importantly, dysregulated miRs (-26a-2-3p, -485-3p) in PsA sera were predicted to strongly (*p*-value < 1 × 10^–325^) regulate the fatty acid biosynthesis and metabolism via fatty acid synthase (FASN) suppression (Figure 2B), suggesting their important role in lipid synthesis. 

### 3.3. Associations between miR Expression and Disease Activity of Rheumatic Patients

Many studies have shown that various miRs may be associated with disease activity scores in rheumatic diseases. Due to the fact that there is no ideal index that assesses disease activity in PsA, we analysed several clinical indexes that can be used to verify disease activity. In our study, ASDAS-CRP positively correlated with mir-92b-5p (rho = −0.58, *p* = 0.03). PsA disease activity measured with DAS28-CRP positively correlated with miR-26a-2-3p (rho = 0.4; *p* < 0.05) and negatively with let-7d-5p (rho = −0.58; *p* = 0.03). Moreover, CPDAI negatively correlated with miR-485-3p (rho = −0.51; *p* < 0.05) and let7d-3p (rho = −0.76; *p* < 0.0005 (Figure 3A–E). Surprisingly, we did not find any relationship between disease activity in RA and miR expression. In AS, ASDAS-CRP positively correlated with miR-485-3p (rho = 0.4; *p* < 0.05) but negatively with let-7d-5p (rho = −0.5; *p* = 0.01) and miR-92b-5p (rho = −0.44, *p* < 0.05) (Figure 3F–H). 

### 3.4. Correlations between Altered Lipid Levels, Fasting Glucose, miRs, and Cytokines in PsA, AS, and RA

Similar to many studies demonstrating that PsA is associated with a higher frequency of metabolic syndrome, hypertension, dyslipidaemia, obesity, and type II diabetes [19], we also observed hypertriglyceridaemia in lipid profile and elevated fasting glucose concentration in our group of PsA patients. We found several correlations between miR expression and cytokine levels corresponding with altered lipid profiles (Table 1, Figure 4 and Appendix A). 

Indeed, miR-10b-5p negatively correlated with TG (rho = −0.73; *p* = 0.003), and miR-146b-3p positively correlated with TG concentrations (rho = 0.54; *p* < 0.05) (Figure 4A,B) in PsA. Additionally, miR-26a-2-3p revealed positive correlations with TC (rho = 0.5; *p* < 0.05) and TG levels (rho = 0.5; *p* < 0.05) (Figure 4C,D). In PsA, fasting glucose level positively correlated both with miR-92b-3p (rho = 0.6; *p* = 0.03) and miR-223-5p (rho = 0.5; *p* < 0.05) (Figure 4E,F). Moreover, expression of miR-223-5p correlated with ESR (rho = 0.65; *p* < 0.001) (Appendix A). In RA, we found mixed dyslipidaemia in lipid profile (Table 1). Similar to PsA, TG levels also positively correlated with miR-26a-2-3p expression (rho = 0.69; *p* < 0.05) (Figure 4G) in RA patients. In addition, we observed a positive correlation between IL-21 and LDL (rho = 0.71; *p* < 0.001) and TC (rho = 0.75; *p* = 0.001) concentrations, respectively (Appendix A), as well as between TNF and non-HDL (rho = 0.97; *p* < 0.05) (Appendix A). In AS, no specific lipid profile was observed.

### 3.5. ROC Curve Analysis of Selected miRs and Cytokines in Distinguishing between PsA, RA, and AS

In linear regression models for RA and correlations for PsA and AS, three different patterns of pro-inflammatory miRs and cytokines were found in PsA, AS, and RA (Figure 4 and Table 3). 

Different pro-inflammatory phenotypes were observed in RA (coefficients with non-normal ranges were excluded). In the linear regression model, DAS28-CRP interchangeability was associated negatively with expressions of miR-222-5p (56%) and miR-92b-3p (44%) (Table 3). 

In ROC analysis, in SpAs expression of miR-26a-2-3p was highly sensitive in diagnosing PsA (78% sensitivity; 72% specificity) and miR-485-3p presented higher specificity for AS (68% sensitivity; 86% specificity) (Figure 5 and Appendix A). In contrast, diagnosis of RA was associated with dominant expressions of let-7d-5p (60% sensitivity; 96% specificity) and miR-146b-3p (80% sensitivity; 55% specificity) (Figure 5 and Appendix A).

### 3.6. Associations between miR Expression and Cytokines Concentration in Serum of Rheumatic Patients

It is well known that miRs show powerful regulatory action on cytokines production; therefore, in the next step, we decided to analyse a possible relationship between previously selected circulating miRs and cytokines involved in the pathogenesis of rheumatic diseases. Surprisingly, we did not observe any correlation between the expression of miRs and the concentration of cytokines in PsA patients. In the RA group, let-7d-5p expression was positively correlated with IL-21 concentration (Appendix A). Additionally, in the same group of patients, there was an inverse correlation between miR-223-5p expression and IL-6 serum level and a positive correlation between miR-92b-5p expression and IL-6 serum level (Appendix A). In addition, let-7d-5p expression and IL-17AF concentration were positively correlated in samples obtained from the AS group of patients (Appendix A). Negative correlations between the expression of miR-485-5p and TNF concentration, as well as between miR-485-5p and IL-21, were observed in AS (Appendix A).

## 4. Discussion

In this study, we demonstrated selected miR profile analysis of PsA, AS, and RA patients. Based on the probability model, we identified disease-specific miR candidates that are known to play important roles in lipid metabolism, pro-inflammatory cytokine production, and disease activity. Indeed, our comprehensive research showed that lipid metabolism and systemic inflammation in SpAs and RA are connected but regulated by different miRs. Additionally, these selected miR profiles can be useful in improving differential diagnostics of patients with early arthritis.

miRs participate in various biological processes and pathogenesis of many diseases. Having been stably detected in body fluids and minimally invasive, the importance of miRs has been recognised over the past few years [20]. In addition, an increasing number of studies demonstrated the key role of miRs as disease-specific biomarkers in autoimmune rheumatic diseases [20,21,22]. In our study, we identified miRs which were altered in PsA (miR-10b-5p, miR-26a-2-3p, miR-485-3p); in AS (miR-10b-5p, miR-26a-2-3p, miR-92b-3p, miR-485-3p); in RA (let-7d-5p, miR-92b-3p, miR-146b-3p, miR-485-3p) (Figure 1). Subsequently, using prediction algorithms from different databases, we conducted a KEGG pathway analysis of selected miRs (Figure 2). Dysregulated miR profiles of AS and RA sera were implicated in ECM–receptor interaction due to negative regulation of collagens and integrins (Figure 2A). The miR profiles of PsA sera, and in particular miR-10b-5p identified downstream, target FASN, which is strongly involved in fatty acid regulation (Figure 2B). Additionally, FASN participates in the production of endogenous ligands for PPARα, which modulates pathways connecting metabolism and systemic inflammation [23,24]. 

Circulating miR profiles in PsA and AS patients were also correlated with disease activity scores of DAS28-CRP, ASDAs-CRP, and CPDAI, respectively (Figure 3A–C). Our results showed that miR-26a-2-3b positively correlated with DAS28-CRP, and let7d-5p negatively correlated with DAS28-CRP; miR-485-3p negatively correlated with CPDAI, whereas miR-92b-3p and let7d-5p negatively correlated with ASDAS in PsA and AS (Figure 3A–E). Interestingly, miR expression profiles were different in peripheral arthritis, compared with axial involvement in PsA patients. Surprisingly, previously selected miRs did not correlate with the DAS28 score in RA patients. Similar to our results showing a negative correlation between miR-92b-3p and ASDAS-CRP (Figure 3C) and a positive correlation between miR-146b and TC in PsA patients (Figure 4B), Pelosi et al. also revealed differentially expressed miR-92b-3p and miR-146b-3p in PsA patients [25]. Another study reported that circulating miR-146b-3p was highly expressed in AS and RA patients and positively correlated with inflammatory cytokine levels [26]. These results are in line with our data showing a significantly elevated level of miR-146b-3p in RA patients and an increased level in PsA and AS patients; however, miR-146b-3p did not reach statistical significance in these patients due to the limited sample size (Figure 1).

In our study, we observed several associations between pro-inflammatory miRs and lipid profile alterations, reflecting differences in lipid profiles between PsA, AS, and RA. In the PsA group, we observed higher fasting glucose levels and hypertriglyceridaemia than in AS patients (Table 1). miR-26a-2-3p is a very promising biomarker in PsA due to its role in the development of obesity and dyslipidaemia [27,28], atherosclerosis [29], and systemic inflammation [30,31]. Simultaneously, in our study, miR-26a-2-3p correlated with altered lipid profile (TC and TG levels), fasting glucose concentration (Figure 4C,D,F), and disease activity measured using DAS28-CRP (Figure 3B). This is a unique observation in PsA, linking previously reported hypertriglyceridaemia and disease activity. We found a 6.6-fold increase (*p* = 0.0034) of circulating miR-10b-5p in PsA (Figure 1) and a significant (*p* = 0.04) negative correlation between miR-10b-5p and TG levels (Figure 4A), supporting KEGG enrichment analysis, which revealed FASN to be a target gene for miR-10b-5p. In our research, we also found that in PsA, upregulation of miR-223-5p is positively correlated with CRP, ESR, and fasting glucose levels. As Chuang et al. reported, miR-223-5p is associated with the development of SpAs [32] obesity, insulin resistance, atherosclerosis [33,34], and upregulation of the IL-17 pathway [34]; therefore, it could also represent a potential novel target as a modulator of inflammation and metabolism in PsA. Similar to other studies [35], miR-92b-3p correlated with fasting glucose level (Figure 4E). As the miR-92b-3p is one of the main anti-inflammatory responses [35] and a glucose metabolism modulator [36,37], the correlation reported in this paper could also represent the connection between systemic inflammation and impaired glucose metabolism in PsA. 

Interestingly, our outcomes suggest that anti-inflammatory response through miR-92b-3p and miR-485-3p might be altered in AS patients. Our research revealed a negative correlation between miR-92b-3p and ASDAS-CRP (Figure 3A) and miR-485-3p negatively correlated with IL-6 and IL-21 (Appendix A). Similar to our findings, in current research, miR-92b-3p and miR-485-3p were described as anti-inflammatory mediators [36,37,38], in addition to metabolism regulators. Paradoxically, in AS, we did not observe correlations between miR-92b-3p, miR-485-3p, miR-26a-2-3p, and the metabolic profile observed in PsA. This observation suggests that the same miRs regulate different pathways in PsA and AS. Interestingly, in the RA group, TG levels correlated with the metabolism regulator miR-26a-2-3p. In recent studies, miR-26-a-2-3p expression was associated with the development of dyslipidaemia, obesity, and altered lipid metabolism [39]; therefore, its lipemic effect could potentially explain increased TG levels in the RA group. Conversely, in RA patients, non-HDL cholesterol and TC levels correlated with TNF and IL-21 levels (Appendix A) but not directly with disease activity or miRs. Finally, circulating let-7d-5p was significantly increased in RA but negatively correlated with ASDAS-CRP in AS (rho = −0.5; *p* < 0.001) and DAS28-CRP in PsA (rho = −0.54; *p* < 0.001) (Figure 3C,F). In line with our study, Pivarsci et al. reported downregulation of let-7 in psoriasis [40], whereas expression of let-7d-5p was increased in RA [41]. 

Establishing diagnosis in the stage of ‘early arthritis’ has significant implications for further disease progression and treatment but is considered challenging. ROC curve analysis (Figure 5) shows that miRs could be a promising new diagnostic tool. miR expression pattern analysis revealed high sensitivity and specificity of the miR-26a-2-3p/miR-485-3p tandem in differentiation between PsA and AS. miR-26a-2-3p was upregulated in PsA and corresponded with disease activity, thus supporting its use in diagnostics [42,43]. However, the extent of the role of miR-485-3p in AS requires further studies. Additionally, let-7d-5p/miR-146b-3p tandem surpasses ACPA and RF [44] in the differentiation of RA from other forms of arthritis or HC. In many studies, including ours, the levels of miR-146b-3p [20] and let-7d-5p were associated with the risk of developing RA [41]. 

The novelty of our research lies in the finding that, in PsA, a shared regulatory pathway exists between obesity, lipid, and glucose metabolism, and systemic inflammation. Unravelled correlations suggest that in PsA, systemic inflammation is driven by TNF, IL-17, and IL-21 (but not through IL-6) and mediated by expressions of miR-26a-2-3p, miR-92b-3p, miR-10b-5p, and miR-223-5p, which was not observed in AS and RA. Additionally, our data stand in line with other research [45], showing the role of miRs in the regulation of lipid metabolism and opening new lines of evidence for including dietary nutrition plans [46,47] and individual training schedules [48] in the treatment of PsA [47]. Currently, there are no certain serological markers for PsA and AS. Thus, our findings might help to distinguish between these two diseases and RA, suggesting the superiority of miR signature over conventional biomarkers (Figure 5). Overall, these data confirmed that miRs can be used as specific biomarkers that surpass traditional serological diagnoses of rheumatic diseases. 

## 5. Conclusions

Our study demonstrated that different miRs contribute to coordinating metabolic adaptation to systemic inflammation in SpAs and RA. This study highlighted a unique regulatory control point through miR-26a-2-3p, miR-223-5p, miR-10b-5p, and miR-92b-3p pathways, connecting proatherogenic metabolism and disease activity in PsA. Conversely, in RA, our findings suggested that lipid profile alterations in RA are secondary to active inflammation. Finally, our study illustrated the potential role of miR-146b-3p, miR-26a-2-3p, miR-485-3p, and let7d-5p in distinguishing between PsA, AS, and RA. 

## Figures and Tables

**Figure 1 biomedicines-10-00893-f001:**
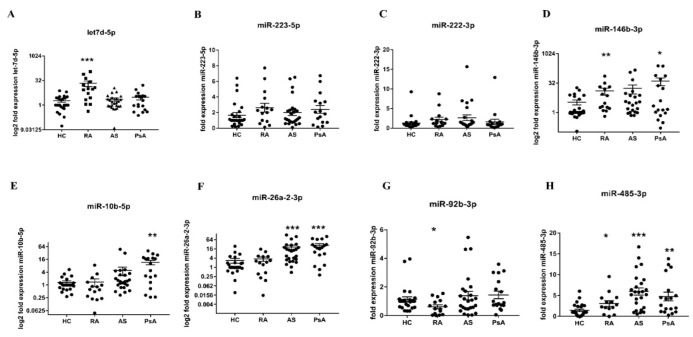
The level of circulating let7d-5p (**A**), miR-223-5p (**B**), miR-222-3p (**C**), miR-146b-3p (**D**), miR-10b-5p (**E**), miR-26a-2-3p (**F**), miR-92b-3p (**G**), and miR-485-3p (**H**). The expression level of sera circulating miRs was measured in HC (*n*  =  15), RA (*n*  =  15), AS (*n* = 25), and PsA (*n* = 18) patients. Results were normalised to the miR-16a internal control. Each symbol represents an individual subject. *p*-values are expressed as follows: 0.05  >  *p*  >  0.01 as *; 0.01  >  *p*  >  0.001 as **; *p*  <  0.001 as ***.

**Figure 2 biomedicines-10-00893-f002:**
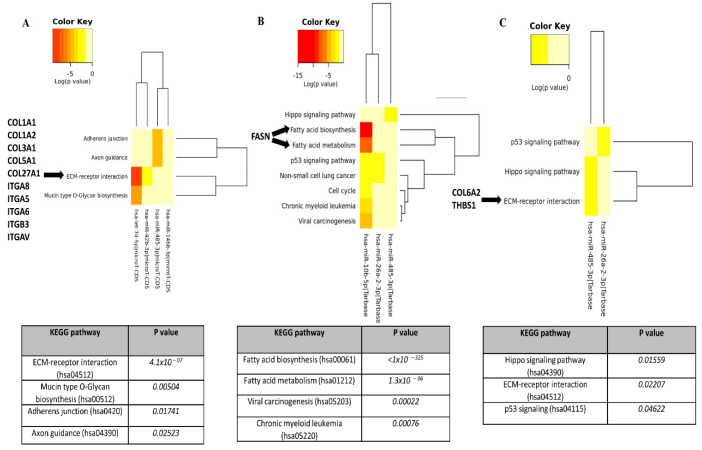
KEGG pathway analysis of significantly changed miRs in RA, PsA, and AS sera. Diana-miPath tool was used to identify specific pathways and their downstream genes, which were negatively regulated by significantly changed miRs in RA (**A**), PsA (**B**), and AS (**C**).

**Figure 3 biomedicines-10-00893-f003:**
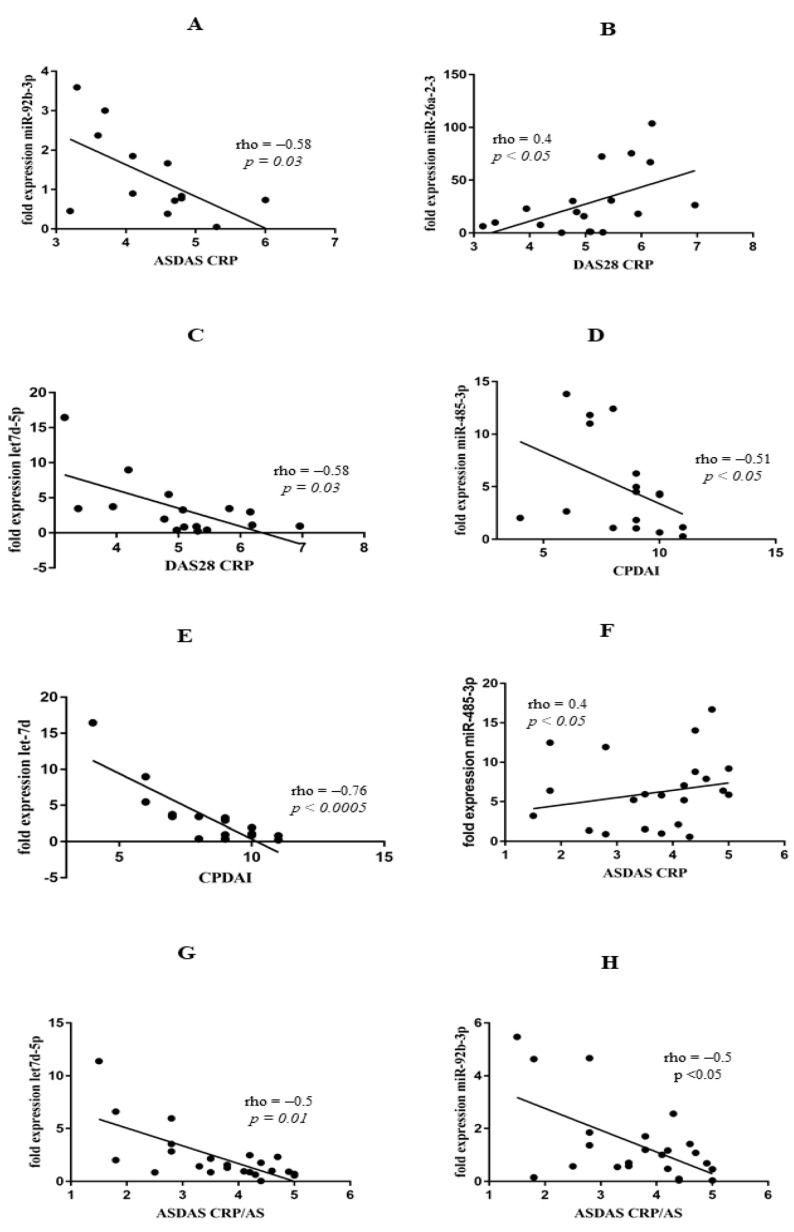
Correlation analyses of miR expression and disease activity scores in PsA group of patients (**A**–**F**) and AS group of patients (**G**,**H**).

**Figure 4 biomedicines-10-00893-f004:**
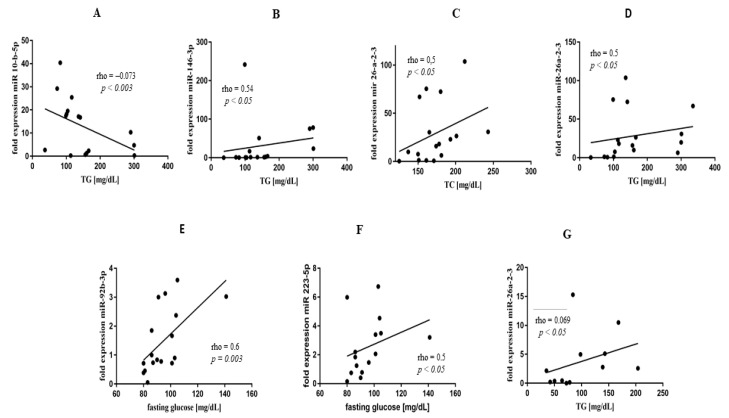
Correlation analyses of miR expression, TG, and glucose concentration in PsA (**A**–**E**) and RA (**F**,**G**) groups of patients.

**Figure 5 biomedicines-10-00893-f005:**
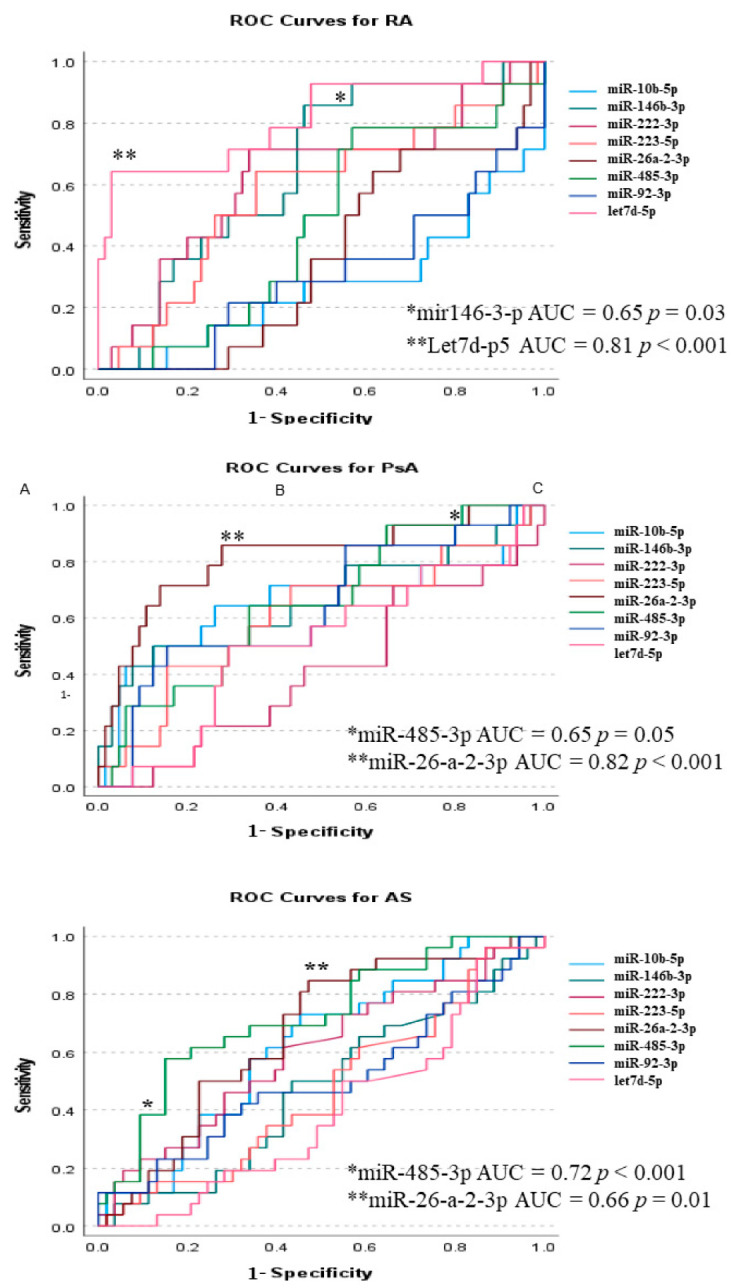
ROC curves for diagnosis of PsA, AS, and RA. *p*-values < 0.05  are expressed as * and *p*-values < 0.001 as **.

**Table 1 biomedicines-10-00893-t001:** Demographic and clinical characteristics of patients.

	RA (*n* = 15)	PsA (*n* = 25)	AS (*n* = 25)	HC (*n* = 25)	*p*
Age (years)	52.4 ± 11.1	44.8 ± 15.4	44.8 ± 13.3	41.9 ± 15.7	0.5
Male/Female	1M/14F	12M/13F	22M/3F	13M/12F	0.2
RF/ACPA positive (%)	75%/87%	0%/0%	0%/0%		<0.001
HLA-B27 (%)	0%	15%	85%		<0.001
Total Cholesterol (mg/dL)	196.7 ± 35.7 (182 IQR 66)	175.4 ± 35.2 (170 IQR 55)	179.1 ± 46.7 (170 IQR 74)		0.96
LDL (mg/dL)	117.1 ± 27.9 (109 IQR 58)	100 ± 26.3 (95 IQR 39.9)	98.9 ± 36.8257 (89 IQR 45)		0.22
HDL (mg/dL)	48.4 ± 12.9 (46 IQR 28)	50.2 ± 13.2 (47 IQR 19)	55.6 ± 21.1 (44 IQR 34)		0.96
Triglycerides (mg/dL)	130.8 ± 78.2 (98 IQR 140)	155 ± 16 (138 IQR 83)	103 ± 45.9 (84 IQR 58)		0.049
Glucose (mg/dL)	100.1 ± 9.6	96.5 ± 14.9	90.6 ± 6.9		0.012
CRP (mg/L)	48.5 ± 37	15 ± 47	11 ± 27.5		0.3
ESR (mm/h)	(38.1 IQR 21.1)	(22.3 IQR 22)	(24 IQR 15.8)		0.026
DAS 28-CRP	5.98 ± 1.1	5.2 ± 0.8	x		0.007
DAPSA	x	38.5 ± 28.4	x		x
CPDAI	x	8 ± 1.6	x		x
ASDAS-CRP	x	x	3.7 ± 1.3		x

**Table 2 biomedicines-10-00893-t002:** Summary of applied treatment.

	RA (*n* = 15)	PsA (*n* = 25)	AS (*n* = 25)
Methotrexate	53%	68%	28%
Sulfasalazine	13%	24%	7%
Glucocorticoids	26%	16%	8%
NSAID’s	30%	24%	72%

**Table 3 biomedicines-10-00893-t003:** Models of linear regression in RA with weights (wi) and adjusted R square of coefficients for DAS28-CRP.

Alternative Models	AIC	Delta AICc	wi	Adjusted R Square
miR-222 + miR-92	1.508	1.392	1.0	0.36
miR-222	2.9	0	0.25	0.373
DAS28-CRP	B	CI	Sig	*p*
miR-222	−0.337	(−0.648: −0.026)	0.564	0.003
miR-92	−0.963	(−1.975:0.049)	0.436	0.05

## Data Availability

Data are available upon request.

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
