# Peer review of "Circulating miRNA Correlates with Lipid Profile and Disease Activity in Psoriatic Arthritis, Rheumatoid Arthritis, and Ankylosing Spondylitis Patients"

_biomedicines, 2022, doi:10.3390/biomedicines10040893_

Round 1
Reviewer 1 Report
The manuscript presented by Krzysztof Bonek et al. title: “Circulating miRNA correlates with lipid profile and disease activity in Psoriatic Arthritis, Rheumatoid Arthritis and Ankylosing Spondylitis patients” is well written, clear, and easy to read. The topic is very interesting (even if per group there are not so many patients) and therefore, it adds clustered information to the subject area of chronic inflammatory conditions such us that tree studied.
In particular, using microRNAs the authors were able to better stratify the patients in base to the specific chronic condition which in turns up today use generic inflammatory blood factors.
Major
1-Please use a different format for the ROC Curve presentation and include in the graph the AUC directly without using a table undernote.
2- Please insert in the discussion this reference and the option to treat those chronic diseases using the nutritional plan option. Please see
- 2019;8(2):116-126. doi: 10.2174/2211536608666181126093903.
- J Funct Morphol Kinesiol. 2021 Jun 24;6(3):58. doi: 10.3390/jfmk6030058.
- Antioxidants (Basel). 2019 Aug 2;8(8):269. doi: 10.3390/antiox8080269.
Minor
- Take out from the author's name Ph.D. title.
Author Response
Thank you for your review. We have revised the manuscript according to your suggestions. Changes to the manuscript were made as follows:
1-Please use a different format for the ROC Curve presentation and include in the graph the AUC directly without using a table undernote.
Graph 5 has been changed according to reviewers suggestions
2- Please insert in the discussion this reference and the option to treat those chronic diseases using the nutritional plan option. Please see
-
2019;8(2):116-126. doi: 10.2174/2211536608666181126093903.
-
J Funct Morphol Kinesiol. 2021 Jun 24;6(3):58. doi: 10.3390/jfmk6030058.
-
Antioxidants (Basel). 2019 Aug 2;8(8):269. doi: 10.3390/antiox8080269.
In lines 352-356 section on potential treatment of PsA with dietary nutrition plan, training was added.
Minor
-
Take out from the author's name Ph.D. title.
It has been changed according to reviewers suggestions

Reviewer 2 Report
The topic of this manuscript is interesting and fits well the scope of Biomedicines. The major limitation of this study is obvious, very limited sample size. As clinical data is used in this manuscript, the reviewer still support the publication of this study subjected to some revisions.
(1) Healthy controls should not be counted as patients (line 83).
(2) How come the sample size information in line 104 and 83 is different?
(3) The information of healthy control should be provided in Table 1 as well.
(4) The gender ratio is very different among the same disease (Table). The authors should discuss whether it will cause problem.
Author Response
Dear Reviewer,
Thank you for your review. We have revised the manuscript according to your suggestions. Changes to the manuscript were made as follows:
(1) Healthy controls should not be counted as patients (line 83).
It has been changed according to reviewers suggestions
(2) How come the sample size information in line 104 and 83 is different?
It is a mistake and has been corrected. not all of the patients presented miRs expression.
(3) The information of healthy control should be provided in Table 1 as well.
The information regarding age and female to male ratio has been added to table 1.
(4) The gender ratio is very different among the same disease (Table). The authors should discuss whether it will cause problem.
This issue has been answered in lines 140-142. In all of the groups (PsA, RA, AS) male to female ratio is similar to populational. RA is significantly more common among women, AS is typically diagnosed in males and in PsA no specific gender distribution has been confirmed.
